# Gut Microbiota—A Future Therapeutic Target for People with Non-Alcoholic Fatty Liver Disease: A Systematic Review

**DOI:** 10.3390/ijms23158307

**Published:** 2022-07-27

**Authors:** Roberta Forlano, Mathuri Sivakumar, Benjamin H. Mullish, Pinelopi Manousou

**Affiliations:** 1Division of Digestive Diseases, Department of Metabolism, Digestion and Reproduction, Imperial College London, London W2 1NY, UK; r.forlano@imperial.ac.uk (R.F.); b.mullish@imperial.ac.uk (B.H.M.); 2Faculty of Medicine, University of Birmingham, Birmingham B15 2TT, UK; ri.svkmr@gmail.com

**Keywords:** NAFLD, NASH, microbiome, probiotic, FMT

## Abstract

Non-alcoholic fatty liver disease (NAFLD) represents an increasing cause of liver disease, affecting one-third of the population worldwide. Despite many medications being in the pipeline to treat the condition, there is still no pharmaceutical agent licensed to treat the disease. As intestinal bacteria play a crucial role in the pathogenesis and progression of liver damage in patients with NAFLD, it has been suggested that manipulating the microbiome may represent a therapeutical option. In this review, we summarise the latest evidence supporting the manipulation of the intestinal microbiome as a potential therapy for treating liver disease in patients with NAFLD.

## 1. Introduction

Non-alcoholic fatty liver disease (NAFLD) represents the most common cause of liver disease in Western countries, due to the pandemic of obesity and metabolic syndrome [1]. Overall, NAFLD includes a wide range of histological features, ranging from simple steatosis (Non-alcoholic fatty liver, NAFL) to non-alcoholic steatohepatitis (NASH), with a variable degree of fibrosis up to cirrhosis and the risk of developing hepatocellular carcinoma [2,3]. Approximately 30% of patients with NAFL develop NASH—histologically characterised by hepatocyte injury and inflammation [4,5]. The main cause of mortality in this population is cardiovascular disease, followed by extra-hepatic malignancy and liver-related events, with the fibrosis stage being the main prognostic factor for all causes of mortality [6,7,8].

The prevalence of NAFLD is increasing in the general population, with an overall prevalence estimated at 29.8% [4,9]. In terms of risk factors, NAFLD is closely associated with a high sugar and fat intake diet, a sedentary lifestyle, and other features of metabolic syndrome: obesity, type 2 diabetes mellitus (T2DM), and dyslipidaemia. Up to 70–80% of individuals with NAFLD may have insulin resistance (IR) or metabolic syndrome [10]. In Asia and the Pacific, NAFLD prevalence has increased at a greater rate than in Western countries, mirroring the rapid lifestyle changes in these regions as well as a different underlying genetic predisposition to metabolic conditions [3,6].

Clinically, as patients with NAFLD often remain asymptomatic, the condition is often missed or diagnosed when progression to advanced stages has occurred. Otherwise, NAFLD is still frequently diagnosed incidentally with abnormal liver blood tests and/or imaging [10,11,12]. Nevertheless, the European Association for the Study of the Liver (EASL) recommends that patients with features of metabolic syndrome should be screened for NAFLD as well as for the presence of fibrosis, using non-invasive markers.

The clinical management of this population is mainly based on weight loss, achieved through diet and physical exercise for the majority of the patients, while bariatric surgery is considered only for those with morbid obesity [7,11]. Despite many promising molecules being in the pipeline for drug development, there is currently no licensed pharmacological therapy for NAFLD [7,11]. As such, there is a lot of interest in novel therapeutic approaches. Among others, manipulating the gut microbiome has become an intriguing target for treating the condition in these patients. This interest has arisen from growing evidence which shows the link between perturbation of the gut microbiome and NAFLD pathophysiology, while specific microbiome compositional signatures have been associated with the severity of liver disease found in these patients [13,14,15]. In this review, we aim to provide an update on the evidence behind the use of different approaches to manipulate the microbiome, for treating patients with NAFLD [16].

## 2. Methods

The aim of this review was to provide an update on the latest evidence supporting the manipulation of the gut microbiome as a potential therapy in patients with NAFLD. A search was undertaken on MEDLINE, PubMed, and Embase for articles—published up to June 2022—containing keywords associated with NAFLD treatments targeted at gut microbiota modulation with evidence from both animal and human studies. Search terms included variations of the following: “NAFLD”, “NASH”, “fatty liver”, “microbiome”, “probiotics”, “prebiotics”, “faecal microbiota transplant/FMT”, “therapy”, “antibiotics”, “diet”, and the search terms were combined in appropriate combinations to generate a list of titles. The search was restricted to English-language papers only. Titles relevant to this review were identified, and abstracts and full texts were screened for applicability. Bibliographies of selected articles were manually screened for additional relevant articles missed in the initial search (Figure 1). The work was compliant with the PRISMA guidelines.

## 3. Results and Discussion

### 3.1. Definition of Intestinal Microbiota

The term “gut microbiota” refers to the microorganism community residing in the intestinal lumen, while the term “gut microbiome” refers to the entire ecological habitat, including the microorganisms as well as their genomes and the surrounding environmental conditions [17]. The adult gut microbiota includes an average 10^13^ bacterial cells, resulting from more than 250 different species of bacteria, fungi, viruses, and archaea [18]. The human intestinal microbiota is mainly composed of bacteria from the *Firmicutes* (60 to 80%), the *Bacteroidetes* (20 to 40%), the *Proteobacteria,* and the *Acinetobacteria* phyla, with high variability among individuals [19,20]. Overall, a wide range of factors may influence the composition and functionality of the gut microbiome, including environmental, immunological, or host factors, as well as alteration in bile flow, gastric pH, or intestinal dysmotility. However, although gut microbiota composition can be modulated by such factors, this is relatively stable in the long term [21]. Of note, the relationship between the host and the gut microbiota is symbiotic and plays a crucial role in modulating the health status. The term ‘dysbiosis’ has sometimes been used to refer to a perturbation of the gut microbiota compared to the ‘normal’ or ‘healthy’ state, although this is an imperfect term since defining the normal, healthy microbiota is itself an area of ongoing debate [20]. Nevertheless, a range of dysbiotic microbiome ‘signatures’ have been associated with a variety of disease states.

### 3.2. The Role of the Intestinal Microbiota in the Pathogenesis of NAFLD

#### 3.2.1. Microbiome Composition

Over the last decade, there has been a growing body of evidence linking the presence of intestinal dysbiosis to the pathogenesis of human liver disease, with a primary focus on metabolic diseases, including NAFLD. Preliminary studies associated NASH with small intestinal bacterial overgrowth in human subjects [22]. Further animal experiments involving the manipulation of the gut microbiome offered then the strongest evidence, supporting the role of dysbiosis in NAFLD. A specific microbiome composition was associated with increased intestinal energy harvest from the diet in obese mice. Interestingly, this trait was shown to be transmissible to lean, germ-free mice via microbiome transfer [23]. Furthermore, IR per se could be ameliorated after the administration of antibiotics [24]. In human studies, when obese men with metabolic syndrome received FMT from lean donors, they showed a significant improvement in IR and an enrichment in their stool of butyrate-producing intestinal microbiota [25].

Over the last years, there has been an explosion of studies exploring modifications in the microbiome and their association with liver disease in NAFLD. A summary of the main changes described in NAFLD is summarised in Table 1. Overall, at the phylum level, an increased abundance of *Proteobacteria* and *Firmicutes*—as well as a reduction in *Bacteroidetes* and *Prevotellaceae*—has been noted in the gut microbiome of patients with NAFLD compared to healthy controls [26,27,28,29]. Notably, the majority of the studies have focused on comparing healthy controls vs. patients with NASH or with simple steatosis, as well as comparing different grades of steatosis. It should also be noted that studies comparing the bacterial taxonomic composition of patients with NAFLD vs. those with NASH produced variable and even contradictory findings, as a result of differences in the cohorts analysed and in the methods used to assess liver disease [30]. Unfortunately, there is only small evidence exploring specific changes in gut microbiota with regards to fibrosis stage in NAFLD, despite this being the main predictor factor in these patients. Furthermore, disentangling the microbial signatures from another co-existing metabolic disease from NAFLD may be challenging [27].

#### 3.2.2. Short-Chain Fatty Acids

Short-chain fatty acids (SCFA)—such as acetic, propionic, and butyric acid—play a crucial role in modulating the interaction between the host and the gut microbiota. Specifically, SCFA are major products of fermentation of undigested carbohydrates (and amino acids) by gut microorganisms up to a daily production of 50–100 mmol/l [31]. The SCFA influence the energetic metabolism, the immune response, and the expansion of the adipose tissue [32]. Many of the effects of the SCFA are mediated via G-protein coupled receptors (GPCRs), which are mainly expressed in the immune system cells, the adipocytes, and the intestinal endocrine cells. Within the intestine, SCFA act on GPCRs, slowing gastric emptying, intestinal transit, and nutrient absorption [33]. Moreover, specific SCFA, such as butyrate, might also suppress inflammation directly as a result of their interaction with T regulatory cells in the intestinal mucosa [34,35].

Interestingly, SCFA were able to reduce the amount of hepatic steatosis, via modulating fatty acid synthetase activity and hepatic lipid synthesis in mice fed with a high-fat diet. In the same model, there was also a two-fold increase in hepatic lipid oxidation in the SCFA-fed mice, mainly due to an enhanced lipid oxidative state [36]. Despite clear results arising from animal models, the role of SCFA in altering the energy harvest has been less elucidated in humans. An early study reported a lower faecal energy excretion in those with obesity when compared with lean ones [37]. Among others, Bacteroidetes are the main contributors to the production of SCFA, with changes in their abundance impacting the level of SCFA. Specifically, it has been demonstrated that a 20% decrease in faecal Bacteroidetes and a correspondent increase in Firmicutes translates into a 150 kcal increase in energy harvest from the diet [23,38]. Of note, such functional change in the microbiota composition can occur after a few days of overeating, hinting at the presence of a dynamic response with caloric intake. On a similar note, another study including adults with NAFLD showed an association between the presence of NASH and an increased percentage of *Firmicutes* vs. a reduced percentage of *Bacteroides*, after adjusting for BMI and dietary fat intake [39]. Not only microbiota but also diet may influence the production of SCFA. Specifically, it is well known that dietary fibres represent an important source of SCFA. Moreover, high-fibre diets may promote the Bacteroidetes phylum, *Prevotella*, whereas high-fat diets reduce diversity and promote Firmicute growth [40].

#### 3.2.3. Bile Acids

Bile acids (BAs) are potent “digestive surfactants” that promote the absorption of lipids, including fat-soluble vitamins. Moreover, BAs are involved in the primary pathway for the metabolism of cholesterol and account for ~50% of its daily turnover [41]. BAs are mainly synthetised in the liver, resulting from the conversion of cholesterol into more water-soluble compounds [42]; BAs are then secreted into the hepatic canaliculi and stored in the gallbladder. After excretion and the digestive process, about 95% of the BAs are re-absorbed from the terminal ileum, while only 5% reach the colon. In the colon, the remaining fraction of BAs is passively reabsorbed after modifications, i.e., deconjugation and oxidation. The intestinal microbiota is actively involved in modulating the pool of circulating and excreted BAs, which in turn participate actively in hydrolysis and dehydrogenation reactions [43]. Overall, BAs display several functions as they are involved not only in the digestion and absorption of lipids, but they also act as signalling molecules modulating the metabolism of glucose and lipids through the farnesoid X receptor (FXR) and the C protein-coupled bile acid receptor TGR5 [44]. In the liver, FXR activation results in the downregulation of free fatty acid (FFA) synthesis and de novo lipogenesis [45]. FXR is also involved in carbohydrate metabolism, as this regulates hepatic gluconeogenesis, and prevents hepatic inflammation [46].

An increased level of BAs has been widely demonstrated in liver tissue [47], plasma [47,48], and faeces [48] in patients with NASH. There is unanimous consensus that higher levels of serum BAs in patients with NASH and NAFL are mainly driven by increased levels of conjugated BAs, while evidence on secondary BAs is still conflicting [49]. A large body of work has demonstrated a dysregulation of BAs metabolism in patients with NASH, including elevated primary conjugated BAs, decreased levels of specific secondary BAs, and alteration of excreted BAs [47,48,50]. Moreover, the expression of BAs transporters also seems to be impacted in patients with NASH or simple steatosis [51,52]. Specifically, the concentrations of cholic, chenodeoxycholic, and deoxycholic acids were significantly increased in the liver of patients with NASH compared to controls [48]. Moreover, cholic acid has been strongly associated with inflammatory markers, with deoxycholic acid showing an opposite trend [48]. Interestingly, a recent study suggested that there might be a specific trend in taurine-conjugated vs. glycine-conjugated BAs, with the first being elevated and the latter suppressed in patients with NASH [53]. Nevertheless, it should be noted that many studies have not accounted for confounding factors such as obesity and insulin resistance, which exert an independent influence on BAs metabolism.

#### 3.2.4. Other Gut-Derived Metabolites

Another postulated mechanism linking the microbiome to NAFLD is its effects on the stimulation of adipose tissue. Specifically, disturbances of the microbiota can result in changes in the production of the intestinal form of fasting-induced adipocyte factor (FIAF). FIAF is a secreted protein which inhibits lipoprotein lipase (LPL) in several extra-intestinal sites, such as white adipose tissue, brown adipose tissue, muscles, and hepatocytes [54]. Inhibiting intestinal FIAF has been linked to increased lipolysis in the adipose tissue and to reduced fatty acid oxidation in the muscles [54]. In the liver, FIAF inhibition results in the activation of lipogenic enzymes and increased fat accumulation [55,56].

Finally, several studies have also demonstrated that the gut microbiota may influence host metabolism in NASH, following an augmented production of dietary ethanol. An early study linked dysbiosis with increased production of ethanol from the intestine; for example, 1 gr of Escherichia coli was able to produce 0.8 gr of ethanol per hour in anaerobic conditions [57]. Additionally, Proteobacteria—a phylum which includes alcohol-producing bacteria—were found to be substantially increased in the gut of patients with NASH [58]. Interestingly, ethanol per se may contribute to liver injury by increasing intestinal permeability and lipopolysaccharide (LPS) levels in the portal circulation, ultimately triggering inflammation [59]. Furthermore, the gut microbiome may elicit the inflammatory response in the hepatocytes and macrophages directly via increased flux of tryptophan metabolites through the portal system [60]. Some other molecules, i.e., ethanol and choline-related metabolites, have also been described as contributors to the development of NAFLD [61].

Given the recent exploration of the role of the microbiota in NAFLD, there has been an increasing interest in evaluating the manipulation of the intestinal microbiome as a potential treatment option for patients with NAFLD. In this sense, different strategies have been investigated, including the use of nondigestible prebiotics, probiotics, and symbiotics [62].

#### 3.2.5. Effect of Microbiome on Insulin Resistance and Adipose Tissue

Several studies have suggested that the gut microbiome may modulate insulin resistance and adipose tissue. Specifically, metabolic inflammation represents the most important player in the way intestinal bacteria have to modulate energy homeostasis. A 40% reduction in microbiome diversity was associated with low-grade systemic inflammation and adipokines production, mainly mediated by LPS production [63]. Similarly, other bacterial products may elicit inflammation and insulin resistance, activating the toll-like receptor (TLR) pathway [64]. The SCFA production also has a direct effect on insulin resistance, as butyrate and propionate may modulate gluconeogenesis and de novo lipogenesis [65]. Moreover, results from the animal model have shown how intestinal bacteria may influence adipose tissue differentiation, with some species favouring brown over white adipose tissue [66,67].

Interestingly, by modifying a dietary pattern, it is possible to change the effect that the microbiome has on insulin resistance. It has been demonstrated that calorie restriction increases *Lactobacillaceae* and reduces *Firmicutes* and *Bacteroidaceae*. Interestingly, a regime of every-other-day fasting led to similar changes [68]. Among others, *Akkermansia muciniphila* appears to be one of the species mediating the beneficial effects of calorie restriction, as it reduces adiposity and increases insulin resistance [69].

### 3.3. Probiotics, Prebiotics, and Synbiotics Use in NAFLD

Probiotics are live bacteria or yeasts that confer health benefits, generally by improving or restoring a healthy intestinal microbiome. Moreover, probiotics are also known to improve intestinal barrier integrity and to have anti-inflammatory effects [70]. Commonly-used probiotic strains include bacteria from the genus *Bifidobacterium* and *Lactobacilli* [71]. Prebiotics are non-digestible food components that promote the growth and survival of beneficial bacteria via intestinal fermentation and these include inulin, raw oats, pectin, and other non-digestible and unrefined carbohydrates [71]. The term ‘synbiotic’ refers to the combined effect of probiotic and compatible prebiotic in a single product with the aim of promoting the survival of such a specific live bacterial strain through the gastrointestinal tract. The majority of prebiotics used in synbiotic supplements include oligosaccharides such as fructo-oligosaccharide, xylose-oligosaccharide, and inulin [71]. Whilst typically used in diarrheal diseases, irritable bowel syndrome, and inflammatory bowel disease, the use of pre-, pro-, and synbiotics has been explored in other areas such as obesity and NAFLD [72,73].

Several human and animal studies have documented that the administration of probiotics can improve hepatic steatosis. Specifically, oral probiotic supplementation with *Lactobacillus fermentum* ameliorated hepatic steatosis, oxidative damage, and inflammatory markers (TNF-α, IL-18) in animal models of diet-induced NAFLD [74,75]. Similarly, a combination of *Bifidobacterium infantis, Lactobacillus acidopilus,* and *Bacillus cereus* led to an improvement in hepatic steatosis and inflammatory infiltration, liver enzymes, serum LPS, and inflammatory cytokines in a high-fat high-sugar diet model. The probiotic supplementation also resulted in a higher abundance of *E. coli* and *Enterococcus* and decreased anaerobes such as *Lactobacillus*, *Bacteroides,* and *Bifidobacteria* [76]. Konda et al. assessed the impact of a five-months supplementation with *Saccharomyces cerevisiae* on rats with high-fat diet (HFD)-induced obesity. There was a significant decrease in body weight, HOMA index, liver tissue peroxidation, liver enzymes, and liver steatosis following treatment. These findings suggest the anti-obesity, antioxidant, and anti-steatosis effects of the probiotic [77].

Studies in humans have demonstrated promising results in probiotic supplementation in NAFLD patients. In a randomised, double-blind, placebo-controlled study, Ahn et al. treated 30 NAFLD patients with a combination of six strains (*Lactobacillus acidophilus*, *L. rhamnosus*, *L. paracasei*, *Pediococcus pentosaceus*, *Bifidobacterium lactis*, and *B. breve*) for 12 weeks. In patients who were on probiotics, there was greater weight loss, mainly from the visceral compartment, a greater decrease in Controlled attenuation parameter (CAP) score, and a greater decrease in total cholesterol, triglyceride, and pro-inflammatory tumour necrosis factor (TNF)-α from baseline compared to the placebo group [78]. On a similar note, other clinical trials have observed an improvement in liver enzymes, inflammatory markers (TNF-α, CRP), and serum endotoxin levels in NAFLD patients following treatment with probiotics alone or with synbiotics (*Bifidobacterium longum* with fructo-oligosaccharide in one study; seven strains of *Lactobacillus*, *Streptococcus,* and *Bifidobacterium* with fructo-oligosaccharide in another study) associated with lifestyle modification [79,80,81,82]. Of note, the 24-week administration of synbiotics led to histological improvement, measured as the NAFLD activity score (NAS) reduced from 9.4 at baseline to 3.2 at the end of treatment [79]. Another study including 39 patients with biopsy-proven NASH showed that a one-year treatment with multistrain probiotic was able to reduce hepatic fibrosis and ballooning [83]. One longer-term study, a double-blind phase 2 trial of 104 NAFLD patients in the UK, assessed the synbiotic treatment of 4 g fructo-oligosaccharide with *Bifidobacterium animalis* (subspecies *lactis* BB-12) twice daily for 10–14 months. The authors observed compositional changes in the faecal microbiota—specifically the increased abundance of Bifidobacterium and *Faecalibacterium* species and the reduction of *Oscillibacter* and *Alistipes* species. However, this was not associated with a reduction in liver fat fraction or with improvement in non-invasive markers of fibrosis, such as Enhanced liver fibrosis test (ELF)/NAFLD fibrosis scores. Only weight loss was associated with a reduction in liver fat [84]. Interestingly, preliminary results from another study suggested that administrating synbiotics could improve liver disease in those with lean NAFLD [82].

Taken together, the evidence from the literature suggests that probiotic and synbiotic supplementation may have a role in treating people with NAFLD in addition to diet and lifestyle modification. Probiotic and symbiotic supplementation could prove particularly relevant in those with a smaller window for weight loss, such as those with lean NAFLD. The safety profile of probiotics and synbiotics makes them a good candidate for the implementation of therapies in clinical practice. However, short follow-up and large heterogeneity amongst studies represent an important limitation to previous studies. As such, more long-term studies are required before further conclusions can be drawn.

### 3.4. Antibiotics Use in NAFLD

Multiple studies have explored the use of antibiotics in NAFLD patients, with the intended aim of modifying the intestinal microbiome and therefore reducing circulating bacterial products [85]. In an animal model, the co-administration of polymyxin B and neomycin prevented hepatic lipid accumulation in high-fructose diet mice [86]. On a similar note, another study assessed the combined effect of capsaicin with antibiotics (vancomycin, 100 mg/kg; neomycin, 200 mg/kg; metronidazole, 200 mg/kg; and ampicillin, 200 mg/kg—once a day for the first week of antibiotic treatment, then three times a week for a furtherseven weeks) on the development of obesity and hepatic steatosis in HFD-fed mice. The authors reported a reduction in hepatic triglyceride and cholesterol levels, reduced BMI, and reduced insulin resistance in the treatment groups, indicating the promising synergistic effects of capsaicin and antibiotics in maintaining intestinal homeostasis and in ameliorating insulin resistance [87].

Rifaximin and solithromycin have been investigated in human studies for treating NAFLD [85,88]. Specifically, rifaximin is a non-absorbable antibiotic that has been used to treat hepatic encephalopathy, traveller’s diarrhoea, and irritable bowel syndrome. Rifaximin acts with a broad anti-Gram positive and anti-Gram negative spectrum, the latter producing LPS. In one open-label study, six NASH patients were treated with Solithromycin for 90 days. Reductions in serum ALT and the NAS were observed in all patients at the end of treatment. In a similar study, rifaximin was administered to 42 patients with biopsy-proven NAFLD for 28 days. In this study, the treatment arm showed a significant reduction in the endotoxin level and liver function tests, but not in serum cytokines [89]. Another study observed a significant reduction in endotoxin and IL-10, in addition to ALT and ferritin in both NASH and steatosis patients, following a 28-day course of Rifaximin (1200 mg/daily) [89]. Nevertheless, another pilot study involving 15 patients with NASH did not observe a beneficial effect following a six-week course of 400 mg rifaximin twice daily. No differences were observed in ALT levels or hepatic lipid content (measured using proton nuclear magnetic resonance spectroscopy), from baseline, following the six-week treatment period, whilst at 12 weeks, serum ALT had significantly increased. No consistent differences were observed in the faecal microbiota at a phylum level in patients and significant differences occurred at a genus level but were not universal to all patients [88].

One of the limitations of these studies has been that clinical endpoints were not based on histology, where liver function tests do not reflect the severity of liver disease in NAFLD [90]. Furthermore, the use of these antibiotics in NAFLD has not been assessed in the long term thus far [91]. Further studies are required to assess the safety and the efficacy of antibiotic treatment in modulating disease severity in NAFLD.

### 3.5. Faecal Microbiota Transplantation in NAFLD

Faecal microbiota transplantation (FMT) involves processing stool from healthy screened donors and using this to transfer an entire microbial community from healthy individuals to the gastrointestinal (GI) tract of affected patients, with the rationale that restoration of a pre-morbid microbiome may mitigate the current condition. FMT is an effective therapeutic option for recurrent *Clostridioides difficile* infection and has shown promise as a strategy for the treatment of metabolic disorders. Previous results from animal models demonstrated that transplanting the gut microbiota from lean or obese mice may induce similar phenotypes to that of the host [92,93]. A study by Zhou et al. on mice with HFD-induced steatohepatitis and treated with FMT over eight weeks, observed decreased intrahepatic lipid and proinflammatory cytokine content and increased the production of intestinal butyrate and the expression of tight junction protein zonula occludens-1 (ZO-1) levels [94]. In terms of gut microbiome composition, FMT in mice increased gut microbiota diversity, increased *Bacteroidetes* abundance, and decreased the abundance of *Actinobacteria* and *Firmicutes* [94]. Regarding human studies, the first ‘signal’ of interest in this field was the recognition that healthy lean donor FMT in patients with metabolic syndrome was associated with transient but consistent improvements in peripheral insulin sensitivity [25,95]; furthermore, a recent randomised trial in patients with severe obesity and metabolic syndrome demonstrated that the combination of healthy donor FMT with a low fermentable fibre supplement was surprisingly more effective at improving insulin resistance than the use of a highly fermentable supplement [96].

The role of FMT as a potential therapeutical option for NAFLD specifically has also been explored in humans. There are currently four registered clinical trials on FMT in NAFLD patients registered on “‘https://clinicaltrials.gov/” accessed on 1 June 2022, including three trials in the recruitment stages. One randomised controlled trial on 21 NAFLD patients receiving donor or autologous FMT did not find significant changes in hepatic proton density fat fraction (measured on MRI) at week 2, 6, or 36 following an allogenic FMT. Interestingly, there was a significant improvement in the intestinal permeability atsix weeks after allogenic FMT in the subset of patients with an increased gut leak at baseline [97]. In another trial, three FMT sessions were undertaken at eight-week intervals on 21 patients, with 10 receiving donor transplants and 11 receiving autologous transplants as the control. Interestingly, recipients of FMT in this study showed an enhanced expression of genes involved in the hepatic lipid metabolism, such as ARHGAP18 and serine dehydratase. There was also an improvement in necro-inflammation on histology, but not in the overall NAS score, steatosis grade, or fibrosis stage [98].

## 4. Conclusions

Whilst there are multiple potential routes into gut microbiota modulation in NAFLD patients, the existing studies have not assessed the long-term efficacy of these therapies and have often omitted histological evaluation in humans, which is necessary for determining the therapeutic potential for NAFLD. Probiotic and synbiotic treatment may serve as a low-risk, low-cost adjunct to the management of NAFLD, yet further investigation is required to identify optimal probiotic formulations. The promise of FMT is yet to be determined as we await the results of the trials in the pipeline.

## Figures and Tables

**Figure 1 ijms-23-08307-f001:**
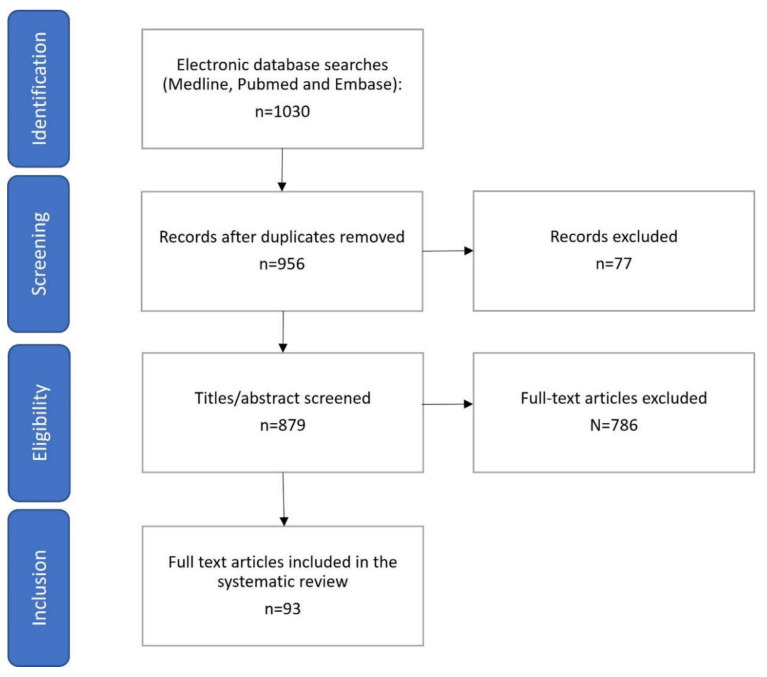
Flowchart of the study for the systematic review.

**Table 1 ijms-23-08307-t001:** Summary of the main alterations of the intestinal microbiota previously described in patients with NAFLD and NASH. The table summarises the main finding from recent studies exploring the association between changes in the microbiome in patients with NAFLD [26,27,28,29].

Disease Severity	Bacterial Microbiota Changes	
NAFLD vs. healthy controls	Phylum	↑ *Proteobacteria*
	Family	↑ *Enterobacteriaceae* ↓ *Rikenellaceae, Rhuminococcaceae*
	Genera	↑ *Escherichia coli*, *Dorea, Peptoniphilus* ↓ *Anaerosporobacter, Coprococcus Eubacterium*, *Faecalibacterium*, *Prevotella*
Severe steatosis or NASH vs. controls or mild steatosis	Phylum	↑ *Fusobacteria*
	Family	↑ *Enterobacteriaceae* ↓ *Prevotellaceae, Clostridiaceae*
	Genera	↑ *Bacteroides, Ruminococcus*, *Shigella*, *Escherichia coli* ↓ *Clostridium*

Abbreviations: NAFLD: non-alcoholic fatty liver disease, NASH: non-alcoholic steatohepatitis.

## Data Availability

Not applicable.

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
