# Peer review of "Gut Microbiota—A Future Therapeutic Target for People with Non-Alcoholic Fatty Liver Disease: A Systematic Review"

_ijms, 2022, doi:10.3390/ijms23158307_

Round 1

Reviewer 1 Report

The aim of the systematic review written by R. Forlano and collaborators was to summarize the latest evidence supporting the manipulation of intestinal microbiome as potential therapy for treating liver disease in patients with NAFLD, since the gut microbiome has become an intriguing target for treating this condition. In order to do this, the authors have made a deep literature search on MEDLINE, PubMed and Embase, using specific keywords.

I think the review is well written, and that the authors stated both the strength and limits of every aspect they analyzed. A part of some minor points to check, overall, I think that the manuscript should be accepted for publication.  

Minor points:

Line 11: You should indicate NAFLD in the abstract too.

Line 13: “there is still no pharmaceutical agent licensed to treat the disease”…I think you can cancel “to treat the diseased”.

Line 14: please substitute “those” with “patients”.

Line 15-16: I think you can cancel “in these patients”.

Line 22: please put a , after “ liver” and remove the bracket.

Line 23: I think would be better to just leave into brackets (NASH) and move “inflammation and ballooned hepatocytes” to line 25 and combine together with “histologically characterized by hepatocyte injury and inflammation”.

Line 41: Please write “recommends”

Line 45: please change “in morbidly obese” with “for those with morbid obesity”

Line 46: please correct “licenced” with “licensed”

Line 49-50: please change “which has shown the” with “showing a”

Line 88: “associated with” is repeated twice.

Line 98-99: please better write the sentence “A specific intestinal microbiome profile was associated with increased intestinal energy harvest from diet in obese mice”

Line 100: you never defined “IR”

Line 102: please write “men with obesity”

Line 105-106: I would write “changes in microbiome composition”

Line 107: please change “changes” with “modifications”. Also, why Table 1.2? There is a Table 1.1 somewhere?

Line 107-108 and Table 1.2: please carefully check what you are writing in the main text and what you are reporting in the table.

Line 111: I would use “mild” instead of “simple”

Line 137-138:  please change “in those with obesity when compared with lean individuals” with “in individuals with obesity when compared with lean ones”

Line 156: please use always the same abbreviation for bile acids, or BA, or BAs.

Line 258: I would specify that NAS score is NAFLD Activity Score. In this way you can remove from line 290.

Line 265: please specify what ELF is.

Line 288: “…LPS producing species”…as it is written it seems is referred to Gram positive species, whilst are the Gram negative the ones producing LPS.

Line 326: please eliminate “a further”

Line 343: please cancel “did not change”

Author Response

Dear reviewer

Thank you very much for your comments. We have made the changes suggested.

            Specifically, with regards to the comment

Line 111: I would use “mild” instead of “simple”.

We decided to keep simple steatosis as the comparison in those studies is NASH vs simple steatosis (non NASH) non per steatosis grade.

Reviewer 2 Report

The paper by Forlano R. and co-workers describes a systematic review process on gut microbiome as a potential target for the treatment of NAFLD.

The topic is of interest and it is suitable for the special issue of IJMS, the review is well performed and the manuscript is accurately written.

Nevertheless, I suggest the following minor modifications to improve the quality of the paper:

- please provide more details related to the review process by improving the representation of the flowchart (I would use the PRISMA software, available at https://www.eshackathon.org/software/PRISMA2020.html );

- page 3, line 107: the Authors refer to a "Table 1.2.". Nevertheless, the manuscript does not include any Table 1.1. Please correct;

- page 1, abstract: the text includes some reiterated words (lines 12-13 "to treat the disease", lines 13-15 "the microbiome"). Please re-write;

- page 1, line 41: recommends

- page 1, line 45: please replace "morbidly obese" with "morbid obesity"

- page 3, line 88: please delete "associated with" (reiteration)

- page 4, lines 160-161: this sentence is not clear. Please revise it

- page 4, line 166: FXR is also involved...

- page 4, line 170: ...of patients with NASH...

- page 5, line 178: ...in of patients with NASH...

- page 5, line 183: ...which have exert...

- page 7, line 276: ...high-fructose diet...

- page 7, line 290: the abbreviation NAS (NAFLD activity score) was previously used in the paper

Author Response

Dear Reviewer

thank you for your comments. We have addressed them and changed the figure. We feel that these changes have strengthened the paper

Reviewer 3 Report

The article needs to be improved. Please discuss in a separatate chpater how microbiota influences insulin resistance and how it increases the fatty acid accummulation in the liver. pleasec consult:Recognizing the Benefits of Pre-/Probiotics in Metabolic Syndrome and Type 2 Diabetes Mellitus Considering the Influence of Akkermansia muciniphila as a Key Gut Bacterium. Microorganisms. 2021; 9(3):618. https://doi.org/10.3390/microorganisms9030618 . Please discuss poteential therapuethic interventions by supplementing with probiotics in order to improve the parameters of fatty liver. please discuss studies where the liver histology was improved after probiotic administration. 

Author Response

Dear reviewer,

thank you very much for your comments. Our point to point reply follows:

The article needs to be improved. Please discuss in a separatate chpater how microbiota influences insulin resistance and how it increases the fatty acid accummulation in the liver. pleasec consult:Recognizing the Benefits of Pre-/Probiotics in Metabolic Syndrome and Type 2 Diabetes Mellitus Considering the Influence of Akkermansia muciniphila as a Key Gut Bacterium. Microorganisms. 2021; 9(3):618. https://doi.org/10.3390/microorganisms9030618 .

                Thank you very much for the comments. The overall effect of microbiome on insulin resistance  is discussed throughout chapter 3.2. However, we agree that this is an important point of discussion and have added another paragraph to the review (from line 219 to 237).

Please discuss poteential therapuethic interventions by supplementing with probiotics in order to improve the parameters of fatty liver.

                By adding another study to the review focused on lean NAFLD (289-291), we implemented the discussion on how the use of probiotics and synbiotics could target patients with small window for lifestyle modifications and weight loss (from 293 to 298).

Please discuss studies where the liver histology was improved after probiotic administration. 

                Thank you very much for this comment. We had already discussed the effect of probiotics on histology and reduction of NAS score (from 277 to 279). We included another study which showed beneficial effect of probiotic supplementation on Histology (from 279 to 282).
